# Genetic Diversity and Structure of Common Carp (*Cyprinus carpio* L.) in the Centre of Carpathian Basin: Implications for Conservation

**DOI:** 10.3390/genes11111268

**Published:** 2020-10-28

**Authors:** Bianka Tóth, Rasoul Khosravi, Mohammad Reza Ashrafzadeh, Zoltán Bagi, Milán Fehér, Péter Bársony, Gyula Kovács, Szilvia Kusza

**Affiliations:** 1Institutes for Agricultural Research and Educational Farm, University of Debrecen, Böszörményi út 138, 4032 Debrecen, Hungary; toth.bianka@agr.unideb.hu (B.T.); bagiz@agr.unideb.hu (Z.B.); 2Department of Natural Resources and Environmental Engineering, School of Agriculture, Shiraz University, Shiraz 71441-65186, Iran; r-khosravi@shirazu.ac.ir; 3Department of Fisheries and Environmental Sciences, Faculty of Natural Resources and Earth Sciences, Shahrekord University, Shahrekord 64165478, Iran; mrashrafzadeh@sku.ac.ir; 4Fish Biology Laboratory, Faculty of Agricultural and Food Sciences and Environmental Management, University of Debrecen, Böszörményi út 138, 4032 Debrecen, Hungary; feherm@agr.unideb.hu (M.F.); barsonp@agr.unideb.hu (P.B.); 5Department of Fish Biology, National Agricultural Research and Innovation Centre, Research Institute for Fisheries and Aquaculture, 5540 Szarvas Anna-liget utca 35, Hungary; kovacs.gyula@haki.naik.hu; 6Animal Genetics Laboratory, Faculty of Agricultural and Food Sciences and Environmental Management, University of Debrecen, Böszörményi út 138, 4032 Debrecen, Hungary

**Keywords:** common carp, genetic variation, Hungary, microsatellites, strains, conservation

## Abstract

Hungary is one of the largest common carp-production countries in Europe and now, there is a large number of local breeds and strains in the country. For proper maintenance of the animal genetic resources, information on their genetic diversity and structure is essential. At present, few data are available on the genetic purity and variability of the Hungarian common carp. In this study, we genetically analyzed 13 strains in Hungary and, in addition, the Amur wild carp, using 12 microsatellite markers. A total of 117 unique alleles were detected in 630 individuals. Low levels of genetic differentiation (F_st_ and Cavalli–Sforza and Edwards distance) were estimated among strains. The AMOVA showed the low but significant level of genetic differentiation among strains (3.79%). Bayesian clustering analysis using STRUCTURE classified the strains into 14 different clusters. The assignment test showed that 93.64% of the individuals could be assigned correctly into their original strain. Overall, our findings can be contributed to complementing scientific knowledge for conservation and management of threatened strains of common carp.

## 1. Introduction

Nowadays many fish species and subspecies are endangered mainly due to habitat loss, overexploitation, pollution, and climate change [1,2]. As a result of highly effective selection programs and robust environmental changes, the risk of losing valuable genetic materials such as old varieties of a certain species, e.g., common carp strains, has increased [3]. Genetic diversity within populations is the basis of their ability to adapt to the changing environment. Lowered genetic diversity might decrease the adaptability and fitness of populations and can result in the increased extinction risk of the populations [4]. The more frequent use of modern strains in new production environments (e.g., intensive fish farming) endangers traditional strains with valuable genetic background. Therefore, obtaining detailed information on genetic diversity and structure within and among populations is crucial for establishing genetic-based conservation efforts for threatened species and subspecies [5], in the case of common carp especially for the effective management and conservation of genetic resources of locally adapted varieties.

The common carp (*Cyprinus carpio* L.), one of the oldest domesticated species for aquaculture [6,7,8], has a high commercial value as a food source for humans, and it has been introduced to various regions of the world [9]. The species has a long cultural history in Europe, dating back to the 16th century. Hungary is the third largest producer of common carp in Europe without Russia [10]. The species has been bred in several local populations in Eurasia, resulting in phenotypic differences in skin color, body shape, body size, and growth rate [11]. Furthermore, common carp strains have been intensively utilized for decades in cross-breeding programs to improve carp stocks [12] and production traits (e.g., better growth rate and feed conversion, etc.). Recent research studies focus on disease resistance to specific pathogens (such as the koi herpesvirus, KHV) [11,13,14,15,16]. More and more studies are using environmental DNA (eDNA) techniques to determine fish community diversity [17].

As a result of the domestication and settling of the species, several local breeds have evolved in Hungary’s natural waters and fish farms [18]. At the end of the 19th century some farmed carp varieties with distinct phenotypic characteristics, e.g., different body shapes, colors, and scales were imported from other localities, such as Aischgründ scaly from Germany, Galician mirror from Czech Republic, Nasic and Polyan mirrors from Croatia, and linear from Poland [19]. Fish production was based on these variants in Hungary, where several varieties adapted to local conditions (e.g., Bikal, Dinnyés, Hortobágy, and Nagyatád), developed by the 1950s [18,19]. In Hungary, the Animal Breeding Authority provides national breed recognition also for those common carp strains or hybrids which completed the defined performance test procedure. After a successful performance test procedure, the authority recognizes the new strain, and the producer can provide proof of origin of that particular common carp strain in the case of sale of genetic material [20,21].

Lehoczky et al. [22] noted that performance testing does not include genetic screening. These tests include evaluation of survival rate, food conversion ratio, growth rate, fat content, dressing yield, abnormalities, and body proportions. The genetic basis of morphological change is an ongoing issue in evolutionary biology, both within and between species, and has recently been a source of debate. There are currently no data available on the genetic purity and variability of all Hungarian strains. However, due to the economic and ecological importance of the species, genetic studies are constantly underway in the world [23,24,25]. Scientists believe these are essential for broodstock and for the genetic improvement of carp species [26,27,28,29].

Previous studies have developed a variety of molecular markers, including microsatellites [23,24,25], which have been shown to be particularly suitable for detecting genetic differences between closely related populations due to their high mutation rates, high polymorphism, abundance, and considerable uniformity throughout the genome and as well as their codominant inheritance [30,31,32,33,34]. Numerous research groups have used microsatellites to characterize local carp strains [1,28,35,36,37,38]. Bártfai et al. [39] analyzed the genetic structure of two Hungarian fish hatcheries (Dinnyes and Attala broodstock) using eight microsatellites and did not find a significant genetic structure between populations. Another study Lehoczky et al. [22], using 12 microsatellites, reported significant genetic differences between six common carp strains in Hungary (Tatai, Biharugrai, Szarvasi, Tiszai, Dunai, Kis-Balatoni).

Hungarian fish farms, similarly to other fish hatcheries of the Carpathian Basin, regard their carp populations as different strains based on their localities and morphology (landraces) [21,40]. In the present study we aimed to evaluate genetic diversity and structure of 13 Hungarian common carp strains using microsatellites. Our results can be a frame for the development of future conservation efforts of the species and help fill gaps in the holistic view of population structure of the species.

## 2. Materials and Methods

### 2.1. Sample Collection and DNA Extraction

A total of 630 samples were collected from 13 Hungarian common carp strains, namely Biharugra scaly (BiS), Biharugra mirror (BiM), Böszörmény mirror (BoM), Hortobágy scaly (HoS), Hortobágy mirror (HoM), Hortobágy wild (HoW), Szarvas-15 (SZ1), Szarvas P3 (SzP), Szeged scaly (SzS), Szeged mirror (SzM), Hajdúszoboszló scaly (HaS), Hajdúszoboszló mirror (HaM), and Tata scaly (TaS) in five hatcheries (Biharugra, Debrecen, Hajdúszoboszló, Hortobágy, and Tiszaszentimre) and the National Agricultural Research and Innovation Centre (NARIC), Research Institute for Fisheries and Aquaculture NAIK HAKI gene bank (National Agricultural Research and Innovation Centre, Szarvas, Hungary). We also collected samples of Amur wild carp (AmW) as outgroup for statistical analysis. Details on the number of samples of each strain, their abbreviations, and coordination of sample collection places for common carp are given in Table 1.

Tissue samples were obtained from fin clips. Approximately 0.5–3 cm of caudal fin clips were cut and preserved in 96% ethanol and stored at −20 °C until further laboratory analysis. The study was performed on the basis of the permit issued by the Hajdú-Bihar Megyei Kormányhivatal (reference number: 15/2019/DE MÁB). Sampling of fin clips was carried out after anaesthetizing the fish in clove oil solution (1 drop of concentrated oil to 1 L of water). Total genomic DNA was isolated from the fin clips using the EZNA Tissue DNA kit (Omega Bio-Tek, Norcross, GA, USA) following the manufacturer’s instructions. Concentration of the isolated gDNA was measured using a Nanodrop 1000 Spectrophotometer. Samples were diluted to 100 ng/μL for further use.

### 2.2. Microsatellite Genotyping

A total of 12 microsatellite loci [41,42] were amplified with fluorescent forward primers (Table 2).

The PCR amplification reactions consisted of 1 µL dNTP (2 mM; Fermentas), 1 µL 10x Gotaq Flexi Buffer (Promega, Madison, WI, USA), 0.7 µL MgCl_2_ (25 mM; Promega, Madison, WI, USA), 0.1 µL reverse and 0.1 µL fluorescent forward primer (100 pmol/µL; Eurofins Genomics, Ebersberg, Germany), 0.05 µL GoTaq Flexi DNA Polymerase (5 U/µL; Promega, Madison, WI, USA), 2 μL genomic DNA (100 pmol/µL), and 5.05 µL sterile water up to a final volume of 10 μL. Thermal cycling conditions for each locus were DNA denaturation at 94 °C for 4 min, followed by 35 cycles of 94 °C for 30 s, annealing temperature (Table 2) for 30 s, and extension at 72 °C for 45 s, and then a final extension of 72 °C for 10 min. Fragment analysis was performed by the laboratory of Biomi Ltd. (Gödöllő, Hungary). The 12 microsatellites were run in four multiplexes (Table 2). The resulting allele is described in PeakSkanner v. 1.0 software (Applied Biosystem, Foster City, CA, USA). We used MICRO-CHECKER [43] to test loci for allele dropout and errors made during the scoring of alleles with ‘stutter’ in our data [43]. We also used FreeNA [44] to estimate the frequency of null alleles (NA; 0.000-0.317) in all loci. The first panel of microsatellites used for the present study consisted of 17 microsatellite loci (Cca24, Cca67, MFW1, MFW2, MFW3, MFW4, MFW6, MFW7, MFW11, MFW13, MFW15, MFW16, MFW17, MFW20, MFW26, MFW29, MFW31). The microsatellite loci showing significantly high rates of null alleles (>0.2) and also uninformative alleles were excluded from further analysis (MFW1, MFW2, MFW16, MFW20, MFW29).

### 2.3. Genetic Diversity

The deviation from the Hardy–Weinberg equilibrium (HWE) and linkage disequilibrium for all locus-strains were calculated using GENEPOP v.4.3 [45,46]. Genetic variation was evaluated according to the basic summary statistics including expected (He) and observed heterozygosity (Ho), mean number of alleles (MNA), allelic richness (Ar), number of private alleles (PA), number of effective alleles (NE), and within-strain inbreeding coefficient (F_is_) with 95% confidence intervals using GenAlEx 6.41 [47], FSTAT v.2.9.3.2 [48], and POPGENE 1.32 [49]. We applied sequential Bonferroni correction [50] to account for the effect of multiple tests by using 1000 dememorization steps, 100 batches, and 1000 iterations per batch.

### 2.4. Population Structure

Pairwise genetic differentiation between strains were calculated using the Wrights fixation index (F_st_) and Cavalli–Sforza and Edwards’ genetic distance (D) [51] using the INA correction method described in FreeNA [44]. The dendrogram was constructed using the genetic distance based on neighbor-joining method [52] with 1000 bootstrap replicates in POPULATION v. 1.2.28 (https://bioinformatics.org/groups/?group_id=84) to explore relationships among the strains [14,53]. Assignment tests were performed using GENECLASS v.2.0 [54] to assign individuals to their strain of origin and translocation events among the strains based on the likelihood of multilocus genotypes using Rannala and Mountain [55] the Monte Carlo resampling of Paetkau et al. [56] with 10,000 simulated individuals and a significance level of 0.05.

The patterns of genetic structure between strains based on allele frequency in microsatellite data were assessed using the analysis of molecular variance (AMOVA) with Arlequin 3.5.2.2 [57]. One thousand random permutations were performed to assess the significance of each pairwise comparison. Strains were grouped for AMOVA in two ways. First, all strains were grouped together, and all loci were considered. Second, the strains were grouped according to their geographical locations (i.e., hatcheries) into six groups (Figure 1).

The pattern of genetic structures among strains was assessed using the Bayesian approach in STRUCTURE 2.3.4 [58], with admixture model and correlated allele frequency [59], a burn-in of 100,000, with 1,000,000 Markov chain-Monte Carlo (MCMC) repetitions and 10 independent runs for each K value (number of clusters, K = 1−15) to check for consistency across runs and identify genetic clusters. The maximum number of clusters was calculated by adding one to the number of strains to allow detection of substructure [60]. The best K genetic cluster was evaluated by calculating ΔK following the Evanno method [61] in STRUCTURE HARVESTER [62]. Then, CLUMPP 1.1.2 [63] and DISTRUCT [64] were used for estimating the highest similarity coefficient over all runs for different values of K and plotting the clustering results.

## 3. Results

### 3.1. Microsatellite Markers

All 14 strains of common carp, in a total of 630 individuals, were successfully screened for 12 microsatellite loci, which were polymorphic in all strains. The analysis by MICRO-CHECKER [43] revealed no indications of stuttering or allele dropout. The frequency of null alleles was generally low ranging from 0.000 to 0.317. Among the 12 loci, evidence of null alleles above the potentially problematic threshold was detected in MFW6, MFW11, MFW13, and MFW26 in some strains (Table 3).

### 3.2. Genetic Diversity

Data for all parameters of genetic diversity for the strains are shown in Table 4. All strains showed deviation from HWE at least at one locus (*p* < 0.01). The mean number of alleles per each strain was 11. The lowest and highest mean number of alleles was observed for Hajdúszoboszló scaly (6.500) and Biharugra mirror (17.580), respectively. A total of 117 private alleles were detected for the 12 polymorphic loci in 630 individuals. The lowest and highest number of private alleles were observed in Szeged mirror and Hortobágy wild, respectively. The mean observed heterozygosity value (H_o_) in the 14 strains was 0.840 with a range of 0.720 in Szarvas-15 to 0.990 in Böszörmény mirror. Additionally, the mean expected heterozygosity value (H_e_) in all the strains was 0.800 with a range of 0.710 in Szarvas-15 to 0.890 in Hortobágy wild (Table 4).

The inbreeding coefficient value varied from −0.250 for Böszörmény mirror to 0.083 for Amur wild carp.

### 3.3. Population Structure

The measure of genetic differentiation (F_st_) ranged from 0.028 (between Biharugra mirror and Hortobágy mirror) to 0.231 (between Szarvas-15 and Hajdúszoboszló scaly; Table 5). The Cavalli–Sforza and Edwards’ genetic distance (D) ranged from 0.347 (between Biharugra mirror and Hortobágy mirror) to 0.723 (between Hajdúszoboszló scaly and Tata scaly; Table 5). The neighbor-joining phylogenetic tree was constructed based on Cavalli–Sforza and Edwards (1967) chord D_c_ genetic distance (Figure 2). The strains studied formed four clusters. Biharugra mirror, Hortobágy mirror, and Biharugra scaly formed one cluster (Cluster 1) connected with Szarvas P3, Tata scaly, Böszörmény mirror, Hajdúszoboszló mirror, Hortobágy scaly, and Hortobágy wild (Cluster 2). These two clusters were connected to cluster 3 (Hajdúszoboszló scaly, Szeged scaly, Szarvas-15, and Szeged mirror). The observed clustering pattern was not consistent with the geographical area of origin. Finally, all samples of Amur wild formed a distinct group connected to the other three previous clusters. Using the population assignment test, 93.64% of individuals were correctly assigned to their strain of origin (Table 4).

AMOVA results (Table 6) revealed that 3.79% of the observed variance occurred among strains, whereas 96.03% was explained by differences within strains. When the analysis was performed after reorganizing the strains in the six geographical groups, the percentage of variation associated to their differentiation was low and non-significant (0.42%, *p* = 0.270), while the difference among strains within geographical groups was higher and significant (3.62%, *p* < 0.001). These results support the conclusion that a high level of genetic variation within strains is biologically typical for domestic common carp in the Carpathian Basin.

The logarithm probabilities Ln *p* (*X*/*K*), using the preliminary STRUCTURE run, related with different numbers of genetic clusters K, calculated from structure analysis of 630 individuals showed the highest value at K = 14, which was followed by K = 2 (Figure 3). Based on the value of K = 14, individuals from various strains were significantly different from each other. Eight of the analyzed strains (HaS, SzS, HaM, BiS, Sz1, SzP, AmW, and BoM) were characterized by very high membership coefficients, and each appeared to represent different gene pools. In contrast, the other strains showed some level of admixture rates.

## 4. Discussion

### 4.1. Genetic Diversity

It is emphasized that the number of used loci and their traits can be strongly affect estimates of genetic parameters [67]. In this study, we evaluated the genetic variation of 12 microsatellite loci within 13 common carp strains in Hungary and Amur wild carp. Based on the findings, all strains differed in at least one locus in Hardy–Weinberg equilibrium, which is also supported by previous studies (e.g., [14]). Several studies [14,35,36] suggested that the deviation from the HWE in the common carp strains may be caused by homozygote excess. However, the deviation from the HWE reported in some strains is caused by heterozygote excess [14]. Overall, European common carp populations are generally characterized by homozygote excess [22,36], therefore this finding may result from different practices of individual fish hatcheries [14].

We detected 117 alleles using 12 microsatellite markers in the common carp strains from Hungary. Tomljanovic et al. [38] reported 148 alleles in Croatian common carp strains using 15 microsatellite loci. Napora-Rutkowski et al. [14] detected 45 alleles within Polish common carp strains based on 15 microsatellites, whereas Lehoczky et al. [22] identified 80 alleles within Hungarian strains based on four loci. However, different results in the number of alleles in different studies may suggest that some loci are more polymorphic.

In accordance with Lehoczky et al. [22] and Hulak et al. [36], we estimated relatively high values of expected heterozygosity within common carp strains (in our study, the values ranged from 0.710 to 0.890). In contrast, Napora-Rutkowski et al. [14] calculated slightly lower expected heterozygosity values ranging from 0.418 to 0.781. In agreement with Napora-Rutkowski et al. [14], the observed heterozygosity values calculated were slightly higher than the expected heterozygosity values.

Based on the inbreeding coefficient, in accordance with Napora-Rutkowski et al. [14], many strains are characterized by a heterozygote excess, thus it can be stated that the possibility of inbreeding deterioration is relatively low. The large number of heterozygotes obtained in our study can also be explained by the high number of alleles, which may also be influenced by the relatively low number of individuals. It has previously been suggested that 50–100 individuals per population are needed for proper estimations of genetic distance measurements and genetic structural indicators [68,69]. However, Hale et al. [70] have recommended that a sample size of about 25–30 individuals per population is required to accurately estimate the microsatellite-based statistics of genetic diversity. Many studies (e.g., [71,72,73]) have demonstrated that increasing the number of individuals in such studies has little benefit in terms of allele frequency and expected heterozygosity. After all, what is important is to give an accurate estimate of allele frequency and diversity, and not to determine all alleles because, for example, some rare alleles are not informative for assessing the genetic diversity or the genetic structure of a population [71,72,73].

### 4.2. Population Structure

In accordance with our results, previous studies reported low levels of genetic differentiation among several strains of common carp in the Czech Republic ([36]; mean F_st_ = 0.183) and France ([35]; mean F_st_ = 0.250). The low F_st_ values can be explained by the inadequate number of elements of the strains, the presence of null alleles, or genotyping error, but this indicator may affect the characteristic features of the strains sampled [14]. The Cavalli–Sforza and Edwards distance supported low levels of genetic differences among the Hungarian strains. The Cavalli–Sforza and Edwards distance have a relatively high sensitivity in detecting genetically similar populations [74]. We found the smallest genetic distance between Biharugra mirror and Hortobágy mirror, whereas the largest genetic difference was observed between Hajdúszoboszló scaly and Tata scaly. This difference can be explained by the geographical distance between these strains (BiM-HoM: geographical distance ≈110 km; HaS-TaS: geographical distance: ≈300 km). While the NJ phylogenetic tree (Figure 2) did not well support the geographical grouping of the common carps, the strains from the Tiszántúl Region (Eastern Hungary) show differences, either phenotypically (mirror, scaly, wild) or geographically. The strains grouped in the first cluster come from the gene bank of the Research Institute for Fisheries and Aquaculture NAIK HAKI (Szarvas, Hungary) and the Debrecen hatchery. This result shows that the strains in the Debrecen hatchery (Szeged scaly and Hajdúszoboszló scaly) can be traced back to the founding individuals from the gene bank (Szarvas-15 and Szarvas P3). Individuals from strains originating from the other four regions show mixing, with the exception of strains from the Biharugra hatchery, which are grouped into a separate cluster. Within the cluster, in the case of the Hortobágy wild we obtained the highest individual private allele (Apr = 24) as well as the highest allele richness (1.890) among these strains. The result of the unique allele richness and the location of the cluster allow us to conclude that Hortobágy wild can be considered as a separate strain, which may be the result of artificial selection. The clustering of strains from geographically close hatcheries suggest occasional exchanges of breeding animals between hatcheries. The strains from the gene bank were grouped into two clusters, but they showed a mixture by strains and phenotype (cluster 1: Szarvas-15 and Szeged mirror, and cluster 2: Tata scaly and Szarvas P3). This result may indicate that the gene bank strains are closer to each other, the genetic distance between them is smaller compared to the strains derived from the hatchery, as similarly found by Lehoczky et al. [22]. Another reason may be that the Szarvas strains were developed from the common carp strains collected from all over Hungary, and that is the reason for the closer relationship of the Szarvas strains to some of the other strains. Furthermore, one possible reason for the Tata scaly and Szarvas P3 located in cluster 2, the reason why the Szarvas P33 scaly carp has been isolated from the Tata homozygote scaly (SSnn) common carp strain with individual selection (oral communication). Thereafter, to consolidate the genetic structure and external characteristics of the new line, inbreeding was used during four consecutive generations, combined with very strict phenotypic selection. The inbred line P33 became the maternal line of Szarvas P31 and P34 hybrids. P3 is the founder population of P33, which is produced from P3 by inbreeding. The Hajdúszoboszló mirror and Böszörmény mirror strains appear in one cluster, which may be due to the fact that in the case of Hajdúszoboszló mirror the Böszörmény mirror strains was used for variety improvement purposes. The Hortobágy mirror and the Hortobágy scaly strains are grouped in two separate clusters, which allows us to conclude that they can be two genetically distinct strains. The first national common carp breeding program led by HAKI back in the 1950s was focused on Hortobágy common carp strains. Hortobágy is the largest fish hatchery and farm in Hungary, with continuous selection and genetic improvement, where strains are maintained under controlled conditions.

In accordance with other studies (e.g., [14,22,67]), over 90% common carps were correctly assigned to their strain of origin. The results of the AMOVA test showed high levels of variance within strains, which suggests high diversity at the level of individuals, but does not support the traditional distinction of strains.

STRUCTURE-based analyses (Figure 3) indicated that the highest ΔK value was obtained when K = 14, where the individuals from various sites differed significantly. This result can be used as evidence for a relatively high genetic diversification of common carp strains in Hungary. Based on the results, we can propose that after their creation, the different strains are genetically pure and kept clean in the case of eight strains (HaS, SzS, HaM, BiS, Sz1, SzP, AmW, and BoM). It is known that a stable polymorphism can be maintained if a heterozygous advantage (overweight) effect exists. In the case of the other six strains (HoS, HoM, HoL, BiM, SzM, TaS), we can already observe genetic evidence for mixing. Well-designed breeding programs can improve this picture, but require careful attention, as uncontrolled mixing can even result in the loss of strains, as has probably already happened, for example, with the Croatian Končanica stock [75].

### 4.3. Conservation Implications

Common carp is the main fish species in pond aquaculture production in Hungary [76], where 33 strains of the species have been identified [18,77]. In addition, almost all Hungarian fish hatcheries distinguish further strains that do not have variety recognition. With few exceptions, the phenotypic difference is negligible between strains [22]. Intensification of breeding and selection programs with high levels of stocks can degrade the genetic basis of the species and lead to the extinction of varieties and the uniformization of populations [36,78]. It is extremely important to genetically describe and preserve our strains, not only for proper maintenance of animal genetic resources, but also in order to be able to choose suitable breeding and selection strategies. Our results on their genetic variability and the relationships between them can be provide a new background for population conservation [79] and breeding programs.

Overall, the microsatellite loci examined in our study proved to be quite effective in characterizing different genetic variabilities within and between the common carp strains. Our findings indicate relatively high values of expected heterozygosity for loci within common carp strains from Hungary. It is known that a stable polymorphism can be maintained if the effect of heterozygous advantage exists. It is also suggested that the applied fish farming practice is able to preserve and possibly improve a certain level of genetic diversity for generations [80,81,82]. In some strains (e.g., Hortobágy scaly, Biharugra scaly, Szeged mirror, and Amur wild carp), based on the inbreeding coefficient, lower heterozygosity values can be seen. These strains require more attention from breeders, as in these cases the risk of inbreeding deterioration can already threaten.

## Figures and Tables

**Figure 1 genes-11-01268-f001:**
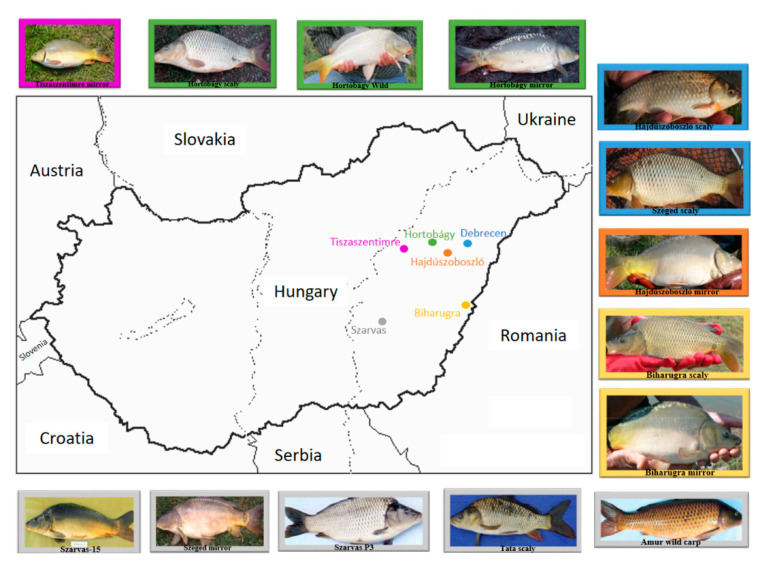
Geographical locations of strains by hatcheries. Cities marked with different colors indicate the hatcheries from which the individuals were collected. Debrecen: Fish Biology Laboratory, Faculty of Agricultural and Food Sciences and Environmental Management, University of Debrecen (Hajdúszoboszló scaly, Szeged scaly); Hortobágy: Hatchery of Hortobágy cPlc. (Hortobágy mirror, Hortobágy scaly, Hortobágy wild); Tiszaszentimre: CLARIAS Agriculture, Producer and Trade Lp. (Böszörmény mirror); Hajdúszoboszló: Bocskai Fishing Ltd. (Hajdúszoboszló mirror); Biharugra: Hatchery of Biharugra Ltd. (Biharugra mirror, Biharugra scaly); Szarvas: Department of Fish Biology, National Agricultural Research and Innovation Centre, Research Institute for Fisheries and Aquaculture (Amur wild carp, Szarvas-15, Szarvas P3, Szeged mirror, Tata scaly).

**Figure 2 genes-11-01268-f002:**
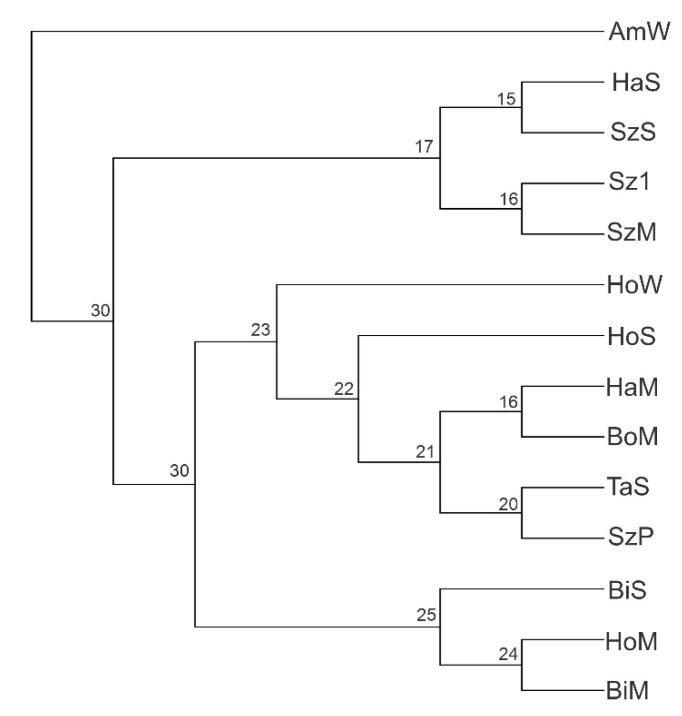
Neighbor-joining tree showing the relationship between 14 common carp strains in Hungary by 12 loci. Bootstrap value obtained from 1000 replicates. Tree generated by TREEVIEW [65] and MEGA version 6.06 [66].

**Figure 3 genes-11-01268-f003:**
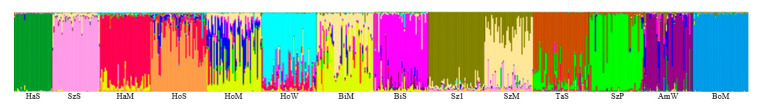
Clustering of individuals by Bayesian algorithm and 12 microsatellites for K = 14 (each cluster is indicated by a unique color). Each vertical column represents one individual, and the separation of the column into 14 colors represents the estimated coefficient of membership to each strain.

**Table 1 genes-11-01268-t001:** Sampling details of common carp (*Cyprinus carpio* L.) strains.

Hungarian Strains	Abbreviation	Sample Collection Place	Sample Collection According to Hatcheries	GPS Coordinate (N: *north latitude*; E: *east longitude*)	Number of Samples (*n*)
Amur wild carp	AmW	Szarvas	Department of Fish Biology, National Agricultural Research and Innovation Centre, Research Institute for Fisheries and Aquaculture	N: 46.51.33.30 E: 20.31.09.56	42
Biharugra scaly	BiS	Biharugra	Hatchery of Biharugra Ltd.	N: 46.56.46.91 E: 21.36.46.96	48
Biharugra mirror	BiM	Biharugra	Hatchery of Biharugra Ltd.	N: 46.56.46.91 E: 21.36.46.96	48
Böszörmény mirror	BoM	Tiszaszentimre	CLARIAS Agriculture, Producer and Trade Lp.	N: 47.29.31.31 E: 20.34.67.56	48
Hortobágy wild	HoW	Hortobágy	Hatchery of Hortobágy cPlc.	N: 47.37.24.13 E: 21.05.17.31	47
Hortobágy scaly	HoS	Hortobágy	Hatchery of Hortobágy cPlc.	N: 47.37.24.13 E: 21.05.17.31	48
Hortobágy mirror	HoM	Hortobágy	Hatchery of Hortobágy cPlc.	N: 47.37.24.13 E: 21.05.17.31	47
Szarvas-15	SZ1	Szarvas	Department of Fish Biology, National Agricultural Research and Innovation Centre, Research Institute for Fisheries and Aquaculture	N: 46.51.33.30 E: 20.31.09.56	48
Szarvas P3	SzP	Szarvas	Department of Fish Biology, National Agricultural Research and Innovation Centre, Research Institute for Fisheries and Aquaculture	N: 46.51.33.30 E: 20.31.09.56	47
Szeged scaly	SzS	Debrecen	Fish Biology Laboratory, Faculty of Agricultural and Food Sciences and Environmental Management, University of Debrecen	N: 47.33.02.91 E: 21.36.19.25	41
Szeged mirror	SzM	Szarvas	Department of Fish Biology, National Agricultural Research and Innovation Centre, Research Institute for Fisheries and Aquaculture	N: 46.51.33.30 E: 20.31.09.56	42
Hajdúszoboszló scaly	HaS	Debrecen	Fish Biology Laboratory, Faculty of Agricultural and Food Sciences and Environmental Management, University of Debrecen	N: 47.33.02.91 E: 21.36.19.25	33
Hajdúszoboszló mirror	HaM	Hajdúszoboszló	Bocskai Fishing Ltd.	N: 47.26.39.68 E:21.20.05.88	43
Tata scaly	TaS	Szarvas	Department of Fish Biology, National Agricultural Research and Innovation Centre, Research Institute for Fisheries and Aquaculture	N: 46.51.33.30 E: 20.31.09.56	48

**Table 2 genes-11-01268-t002:** Information on microsatellite primers used in PCR.

Locus	Forward and Reverse Sequence (5′-3′)	Fluorescent Dye	Annealing Temperature (°C)	Fragment Range (bp)	Multiplex for Fragment Analysis	Reference
Cca24	AAATTTTCAAGACTGGGTGGTT ACAGCAAGATGACAAAATGAGTG	ATTO565	60	210-234	1	[42]
Cca67	GTAGCCCCAAAAGATGTAGCA TGGTCAAGTTCAGAGGCTGTAT	FAM	60	209-299	3	[42]
MFW3	GATCAGAAGGTACAGAGAAG CCTTACAGAAAACCTGTTTGC	ATTO565	58	134-240	1	[41]
MFW4	TCCAAGTCAGTTTAATCACCG GGGAAGCGTTGACAACAAGC	HEX	59	138-253	1	[41]
MFW6	ACCTGATCAATCCCTGGCTC TTGGGACTTTTAAATCACGTTG	FAM	60	158-212	2	[41]
MFW7	GATCTGCAAGCATATCTGTCG ATCTGAACCTGCAGCTCCTC	ATTO550	59	132-152	2	[41]
MFW11	GCATTTGCCTTGATGGTTGTG TCGTCTGGTTTAGAGTGCTGC	ATTO565	60	110-196	2	[41]
MFW13	ATGATGAGAACATTGTTTACAG TGAGAGAACAATGTGGATGAC	HEX	58	192-270	2	[41]
MFW15	CTCCTGTTTTGTTTTGTGAAA GTTCACAAGGTCATTTCCAGC	ATTO550	59	159-283	3	[41]
MFW17	CAGTGAGACGATTACCTTGG GTGAGCAGCCCACATTGAAC	HEX	60	254-312	1	[41]
MFW26	CCCTGAGATAGAAACCACTG CACCATGCTTGGATGCAAAAG	ATTO550	60	151-221	4	[41]
MFW31	CCTTCCTCTGGCCATTCTCAC TACATCGCAGAGAATTCGTAAG	ATTO550	60	283-305	1	[41]

**Table 3 genes-11-01268-t003:** Estimated null allele frequencies in 14 strains of the common carp using 12 microsatellite loci.

Locus/Strain	HaS	SzS	HaM	HoS	HoM	HoW	BiM	BiS	Sz1	SzM	TaS	SzP	AmW	BoM
Cca24	0.087	0.000	0.000	0.001	0.000	0.021	0.000	0.000	0.066	0.000	0.000	0.000	0.083	0.000
MFW3	0.000	0.000	0.000	0.000	0.000	0.000	0.000	0.000	0.000	0.000	0.000	0.000	0.000	0.000
MFW4	0.000	0.000	0.000	0.000	0.000	0.000	0.000	0.000	0.000	0.000	0.067	0.000	0.000	0.000
MFW17	0.000	0.023	0.000	0.000	0.006	0.037	0.000	0.020	0.000	0.000	0.000	0.000	0.000	0.000
MFW31	0.000	0.000	0.000	0.000	0.000	0.000	0.008	0.000	0.000	0.000	0.000	0.000	0.000	0.000
MFW6	0.001	0.192	0.041	0.048	0.069	0.000	0.033	0.085	0.064	0.173	0.040	0.202	0.189	0.000
MFW7	0.000	0.088	0.000	0.050	0.000	0.000	0.000	0.012	0.058	0.000	0.014	0.000	0.000	0.000
MFW11	0.000	0.160	0.000	0.000	0.133	0.237	0.262	0.164	0.000	0.305	0.000	0.000	0.317	0.000
MFW13	0.000	0.184	0.000	0.016	0.011	0.000	0.000	0.154	0.258	0.298	0.011	0.000	0.184	0.000
Cca67	0.000	0.000	0.000	0.044	0.000	0.000	0.000	0.000	0.000	0.000	0.000	0.051	0.000	0.000
MFW15	0.000	0.000	0.000	0.000	0.000	0.000	0.000	0.000	0.000	0.000	0.000	0.000	0.000	0.000
MFW26	0.000	0.000	0.234	0.212	0.000	0.086	0.206	0.192	0.063	0.000	0.007	0.000	0.095	0.000

Taking into account the low frequency of null alleles and the non-significant results of MICRO-CHECKER for all strains, we did not discard any loci from further analyses.

**Table 4 genes-11-01268-t004:** The measures of genetic variation within 14 common carp strains in Hungary using 12 microsatellite loci. MNA, mean number of alleles per strain; Apr, number of private alleles; Ar, mean values of allelic richness; N_e_, mean number of effective alleles; H_o_, observed heterozygosity; H_e_, expected heterozygosity; dHW, number of loci deviating from HWE; F_is_, inbreeding coefficient; assignment test, percent of individuals correctly assigned to their strain of origin.

Strain	MNA	Apr	Ar	Ne	H_o_	H_e_	dHW	F_is_	Assignment Test
HaS	6.500	9	1.670	5.290	0.790	0.730	1	−0.080	100
SzS	8.180	4	1.680	4.270	0.770	0.750	3	−0.027	97
HaM	8.450	2	1.820	5.670	0.920	0.820	2	−0.118	88
HoS	11.080	11	1.860	7.810	0.860	0.860	1	0.002	97
HoM	14.000	4	1.830	6.470	0.900	0.830	2	−0.083	89
HoW	12.250	24	1.890	9.710	0.900	0.890	1	−0.007	93
BiM	17.580	1	1.800	5.240	0.840	0.800	2	−0.051	83
BiS	11.500	8	1.830	7.090	0.810	0.840	3	0.026	91
SZ1	13.000	1	1.740	4.640	0.720	0.710	3	−0.004	97
SzM	9.630	0	1.770	4.210	0.730	0.750	1	0.034	88
TaS	7.900	11	1.820	5.910	0.930	0.820	3	−0.134	95
SzP	11.580	3	1.790	4.960	0.900	0.790	4	−0.140	93
AmW	8.500	17	1.800	5.870	0.740	0.810	1	0.083	100
BoM	13.080	22	1.800	5.360	0.990	0.800	5	−0.250	100

**Table 5 genes-11-01268-t005:** Pairwise F_st_ values (above diagonal) and D_c_ distance (below diagonal) between 14 common carp strains in Hungary based on 12 microsatellite loci.

Strain	HaS	SzS	HaM	HoS	HoM	HoL	BiM	BiS	SZ1	SzM	TaS	SzP	AmW	BoM
HaS	0.000	0.164	0.173	0.158	0.134	0.131	0.155	0.152	0.231	0.188	0.176	0.181	0.147	0.174
SzS	0.565	0.000	0.146	0.137	0.122	0.116	0.135	0.116	0.204	0.114	0.146	0.149	0.131	0.140
HaM	0.702	0.647	0.000	0.041	0.056	0.036	0.066	0.077	0.176	0.096	0.080	0.074	0.112	0.066
HoS	0.702	0.649	0.454	0.000	0.051	0.035	0.071	0.076	0.155	0.106	0.071	0.072	0.102	0.090
HoM	0.601	0.570	0.497	0.479	0.000	0.043	0.028	0.040	0.136	0.067	0.079	0.074	0.089	0.111
HoW	0.652	0.624	0.491	0.476	0.446	0.000	0.053	0.054	0.127	0.085	0.054	0.057	0.078	0.063
BiM	0.605	0.561	0.476	0.518	0.347	0.479	0.000	0.040	0.131	0.047	0.101	0.098	0.084	0.121
BiS	0.651	0.559	0.555	0.548	0.405	0.490	0.400	0.000	0.110	0.067	0.103	0.105	0.077	0.113
SZ1	0.683	0.605	0.652	0.615	0.555	0.595	0.530	0.523	0.000	0.099	0.181	0.196	0.133	0.194
SzM	0.630	0.465	0.547	0.615	0.498	0.585	0.429	0.493	0.437	0.000	0.113	0.131	0.112	0.139
TaS	0.723	0.660	0.513	0.519	0.559	0.527	0.585	0.607	0.666	0.606	0.000	0.039	0.120	0.106
SzP	0.714	0.652	0.496	0.532	0.521	0.539	0.559	0.580	0.683	0.592	0.389	0.000	0.126	0.124
AmW	0.618	0.581	0.622	0.615	0.536	0.564	0.500	0.512	0.563	0.584	0.625	0.612	0.000	0.149
BoM	0.687	0.632	0.498	0.561	0.633	0.538	0.632	0.660	0.700	0.665	0.560	0.600	0.684	0.000

**Table 6 genes-11-01268-t006:** Analysis of molecular variances (AMOVA) among 14 strains of common carp from Hungary.

Number of Groups	Source of Variation	df	Sum of Square	Variance Component	Percentage Variation	*p*
One group	Among strains	13	25.270	0.017	3.790	0.000
Within strains	1264	513.670	0.412	96.030	
Total	1259	538.940	0.429		
Six (geographical) groups	Among groups	5	10.590	0.001	0.420	0.270
Among strains within groups	8	14.680	0.015	3.620	0.000
Within strains	1246	513.67	0.412	95.960	0.000
Total	1259	538.94	0.429

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
