# Peer review of "Genetic Diversity and Structure of Common Carp (Cyprinus carpio L.) in the Centre of Carpathian Basin: Implications for Conservation"

_genes, 2020, doi:10.3390/genes11111268_

Round 1

Reviewer 1 Report

Review of “Genetic diversity and structure of common carp (Cyprinus carpio L.) in the centre of Carpathian Basin: implications for conservation”

In this manuscript, the authors utilized 14 previously published microsatellites markers to assess the genetic diversity of a large sampling number of various carp strains from different fisheries. The authors declare the methods are effective to differentiate strains of carp and I am highly confident the authors have appropriately analyzed their data. The manuscript is easy to read with only minor grammatical errors and typos which I have done my best to address below. While the findings may not be the most surprising (strains of carp determined by breeding history and phenotypically can also be determined genotypically), the methods and results are sound and in combination with other studies, may help guide breeding strategies of fish farms. I believe, however, the manuscript can be improved in several ways. I am not asking for additional studies, just improvements to the writing to better summarize what is known and what this manuscript is adding.

1. There appears to be a wealth of literature that has already genotypically assessed many of these carp strains, some used the same molecular markers used here. Can the authors please describe in more detail how their results compare to these other studies? Did the authors detect any novel or surprising groups of these strains? For instance, line 316 describes that two strains grouped into separate clusters and are therefore genetically distinct strains. Is this the first study to make these conclusions? Given that the NJ tree shows very low bootstrap support, I would be hesitant to believe this conclusion if it contradicts other studies. The impact of this paper would be greatly increased if the authors summarized what was found by others with many more details, rather than the broad strokes used in sections 4.1 and 4.2.

2. The authors have two figures, however, direct references to these figures are only found in the results sections, but section 4.2 has multiple places were direct references would clarify the discussion points.

3. One of the points the authors tried to make (line 158), is that strains grouped according to the geographical locations. This could easily be turned into a figure, showing a map of sampling locations and group id found at each location.

4. Please provide genotyping data on a public repository so that others may expand on this work. There are a number of appropriate places to choose from (Dryad for example) but I would recommend checking with the editor to see what is recommended by Genes.

Minor typos/suggestions:

Line 39: remove comma at “Lowered genetic diversity,”

Line 40, switch “that” to “and”

Line 45, “loosing” to “losing”

Paragraph 37-43 is redundant with 44-51. Combine these two into a single paragraph as they both make the same “crucial” point.

Line 87, “other” to “another”

Table 2: I’m not familiar with using “F” as a DNA base. Is this a typo? Is this meant to show which primer was fluorescently labeled? If that is the case, some of the primer pairs are missing the “F” label.

Methods at line 122 either need to include the final concentration of reagents used, or the amount used. The methods refer to Table 2 for determining which PCR products were multiplexed before the fragment analysis run, however table 2 does not include this information.

Line 171, is there a citation/version for “MICRO-CHECKER”?

Line 204, could the authors please expand on the low bootstrap values observed in Figure 1 and what this may mean? Are they confident about the clusters?

Paragraph 242-250. I’m not familiar with the terms “homozygous surplus” and “heterzygous surplus” and a quick google search and scanning of the articles cited did not clarify what was meant. Are these common terms? Could the authors please clarify and expand on what they mean?

Line 251, 117 alleles across all microsatellite loci? Could the authors be more specific if they mean that with their 12 markers, 117 alleles were detected?

Line 261, rearrange the sentence to be “the observed heterozygosity values calculate were slightly higher…”

Line 270, “acquire” to “required”

Line 300, “it can be concluded” is a strong beginning to the sentence that ends with “may” and there could be other explanations. I would suggest changing “it can be concluded” to something less strong, such as “The clustering of strains from geographically close hatcheries suggest occasional exchanges of breeding animals between hatcheries”.

Author Response

1.

There appears to be a wealth of literature that has already genotypically assessed many of these carp strains, some used the same molecular markers used here. Can the authors please describe in more detail how their results compare to these other studies? Did the authors detect any novel or surprising groups of these strains? For instance, line 316 describes that two strains grouped into separate clusters and are therefore genetically distinct strains. Is this the first study to make these conclusions? Given that the NJ tree shows very low bootstrap support, I would be hesitant to believe this conclusion if it contradicts other studies. The impact of this paper would be greatly increased if the authors summarized what was found by others with many more details, rather than the broad strokes used in sections 4.1 and 4.2.

Answer:

Thanks to her favourable hydrographic characteristics and production traditions, Hungary can be considered as a veritable fish-farming power. Common carp play a primary role in Hungary, since this is the economically most significant fish species. Hungary is the third largest producer of common carp in the European Union. Hungarian hatcheries consider their common carp stocks as separate strains. The number of Hungarian strains is quite high as compared to those registered in the neighbouring countries (at present 33 strains are accepted as i.e. landraces; in addition to these, however, nearly each Hungarian hatcheries also distinguishes further strains). The reason for this is found in production traditions and the national system of agricultural subsidies, since maintainers of the officially accepted strains are entitled to state subsidy. The question therefore arises how well-grounded the biological foundation of the differentiation of these strains really is. Only limited special literature is available on the genetic background of the various Hungarian common carp strains, in spite of the fact that in Hungary common carp not only serve as food but, being a species domesticated a long time ago, also represent an important cultural value. So far the common carp strains studied in our laboratory, bred in the Tiszántúl Region (the area east of the river Tisza) have not been subjected to comprehensive molecular genetic analysis.

In our study we sought to present all publications using microsatellite markers for the characterization of Hungarian strains. Two such works have been published in the early 2000s [22,39], reporting the following results. Bártfai et al. [39] published a study on two Hungarian fish hatcheries (Dinnyes and Attala broodstocks) using 8 microsatellites (MFW4, MFW7, MFW9, MFW13, MFW17, MFW20, MFW26 and MFW31). The results did not show a significant difference between the genetic structures of the two populations: heterozygosity values and allele frequencies were similar. The dendrograms did not show a separate group among the populations. Genotypes from the two populations were compared to those from other fish hatcheries (Böszörmény (n=50), Tata (n=35), Bikal (n=6), Szajol (n=5) and two rivers (Danube (n=4) and Tisza (n=4)) using a limited number of samples. Allele frequencies were similar in the strains, except for the wild carp strains. Lehoczky et al. [22] published a study on the genetic characterization of six Hungarian common carp strains (Tatai, Biharugrai, Szarvasi, Tiszai, Dunai, Kis-Balatoni) using 12 microsatellites (CCA21, MFW3, MFW6, MFW7, MFW9, MFW10, MFW12, MFW13, MFW14, MFW16, MFW23 and MFW29). Their results demonstrated differences between the strains. Their pairwise Fst values (0.013-0.161) were highly significant. The average number of alleles ranged from 3.9 to 8.2. The mean observed heterozygous value (0.557) was lower than the expected heterozygous value (0.700). Based on their results, the Danube tribe was an exception, where the observed heterozygous value (0.764) was higher than the expected heterozygous value (0.602). At five loci, extremely high heterozygous values were observed for the Danube strain. In the two wild strains (Dunai and Tiszai), there was a deviation from the Hardy-Weinberg equilibrium at several loci. The relatively high number of individual alleles and the allele frequencies obtained demonstrated that the individuals were properly assigned to the strains (over 90%).

The publications presented above utilized several microsatellites used in our work (MFW3, MFW4, MFW6, MFW7, MFW13, MFW17, MFW26, MFW31) and characterized some strains discussed in our study (Tatai, Biharugrai, Böszörményi, Szarvasi); in spite of these overlaps, however, there is little similarity with our study, and it would be difficult to carry out direct comparisons based on the published values. Nevertheless, our results often support the observations made by Lehoczky et al. [22] and Bártfai et al. [39], for example as regards He and Ho values, the significant but rather small differences between the genetic compositions of the broodstocks, and the high number of unique alleles.

Our study used higher sample numbers and included more microsatellite markers than did previous molecular genetic studies on Hungarian common carp strains. In our opinion the genetic characterization of the strains is eminently important not only for the appropriate maintenance of animal genetic resources, but also for supporting the choice of adequate breeding and selection strategies. We think that, by our gap-filling study, our results concerning the genetic variability and the interrelationships of Hungarian common carp strains may not only provide a novel background for the protection of Hungarian common carp populations and for breeding programs, but may also extend the existing global knowledge base about this economically important fish species.

  1.  

The authors have two figures, however, direct references to these figures are only found in the results sections, but section 4.2 has multiple places were direct references would clarify the discussion points.

Answer:

Indeed there was no reference to the two Figures in section 4.2, which we supplemented following the Reviewer’s advice, to help easier understanding.

  1. 3.

One of the points the authors tried to make (line 158), is that strains grouped according to the geographical locations. This could easily be turned into a figure, showing a map of sampling locations and group id found at each location.

Answer:

We thank the Reviewer for this suggestion, which we implemented during revision of the manuscript. Table 1 was corrected, and a new column entitled „Sample collection according to hatcheries” was added and drew up a map thereof (Figure 1).

  1.  

Please provide genotyping data on a public repository so that others may expand on this work. There are a number of appropriate places to choose from (Dryad for example) but I would recommend checking with the editor to see what is recommended by Genes.

Answer:

Thank you for your advice; we will upload our raw data on the interface recommended by Genes.

Minor typos/suggestions:

Answer:

Thank you for your comments, suggestions and for pointing out grammatical errors. We hope all of these have been implemented, corrected or modified in the revised manuscript.

9.

Line 204, could the authors please expand on the low bootstrap values observed in Figure 1 and what this may mean? Are they confident about the clusters?

Answer:

We thank the Reviewer for his/her comment on the low bootstrap values. There have been very few reports published on the genetic diversity and the population structure of the Hungarian common carp. That is why we decided that any additional results on the geographical region of the Carpathian Basin may play a gap-filling role. Our results add new elements to our molecular genetic knowledge primarily about the Hungarian common carp strains. We are also convinced that our results can promote the more efficient realization of breeding programs and the protection of the strains, since preservation of valuable genetic materials is only feasible when as much information as possible is available about the given species or variety.

Reviewer 2 Report

Dear authors,

In this article authors study the genetic structure of 14 strains of carp in Hungary by means of microsatellite loci. The manuscript is easy to follow and read and it’s understandable from the view of a non-native English speaker. The methodology is correct and well described. Despite the analysis of the results used is sufficient, from my humble point of view, it could be improved.

In the following lines I will try to explain in the most coherent and constructive way possible on how to do so. The interpretation of the results is correct (although I added some minor suggestions below), the available references were dully cited and the conclusions are pretty much supported. Consequently, I suggest minor corrections, although I encourage authors to follow my advices.

As I said, the workflow of data analysis is pretty much correct, authors have done the whole valid approach for microsatellite analysis. I got, though two suggestions.

The first suggestion would be for authors to employ a more appealing display of the results. In this sense I encourage them to perform a Discriminant Analyses of Principal Components analysis, additional to the NJ tree they’ve performed. Personally, I think this could be easily done with the package adegenet in R (see DAPC) and could be a nice addition/checking to these results.

The second (and related to the first one) one would be for authors to check whether some of the tested loci were to be, in fact, uninformative. If so, some of the loci could be actually adding some noise to the analysis and, if dropped, the clustering reanalysis (both at structure and DAPC) would be much cleaner than it is.

Other minor comments:

L116      Such details are not necessary in a manuscript like this (in my view)

L123-124             Why a pair of "/" in  "/Eurofins Genomics/"

L133      Actually some loci with NA's higher that 0.2 were, in fact, included in further analysis, as indicated at lines 179-180. Please, change this to something like "loci showing significantly different NA proportions were excluded from analysis". Also, indicate here the chosen significance threshold and by which program this was calculated.

L149      The link (http://bioinformatics.org/tryphon/populations) is down. Please provide a valid one.

L159      A table detailing to which geographical location group belongs each strain would be a nice addition, perhaps adding a column to table 1.

L163      Actually, when running structure, I always recommend testing more K values than a priori populations to check the possibility of hidden genetic sub-populations within the studied metapopulation.

L184      Please, when talking about mean alleles, do so in whole numbers; There is no way that any animal has a tenth of an allele

L230      Actually, showing/discussing the second-best K (K=2), even in one liner, could be also illustrative. Please, consider this possibility.

Author Response

  1.  

The first suggestion would be for authors to employ a more appealing display of the results. In this sense I encourage them to perform a Discriminant Analyses of Principal Components analysis, additional to the NJ tree they’ve performed. Personally, I think this could be easily done with the package adegenet in R (see DAPC) and could be a nice addition/checking to these results.

Answer:

Thanks for your comment. We used Principal Coordinates Analysis (PCoA), as an ordination method, due to access to geographic data of individuals. This analysis was performed using GenAlEx 6.41 [47] at the individual level when editing the manuscript. In accordance to STRUCTURE analysis and also NJ tree there is no significant genetic structures between various strains, therefore the figure not included in the manuscript.

First and second components of a principal coordinate analysis (PCoA) of 12 microsatellites.

  1.  

The second (and related to the first one) one would be for authors to check whether some of the tested loci were to be, in fact, uninformative. If so, some of the loci could be actually adding some noise to the analysis and, if dropped, the clustering reanalysis (both at structure and DAPC) would be much cleaner than it is.

Answer:

Thanks for your comment. The first panel of microsatellites used for the present study consisted of 17 loci. Before starting the analysis, we removed loci with high levels of missing data and also uninformative alleles. This reduction of loci resulted in the inclusion of only 12 loci for further analysis. Based on this, we clarified the description in the manuscript.

Other minor comments:

  1.  

L116 Such details are not necessary in a manuscript like this (in my view)

Answer:

We thank the Reviewer’s comment. We wished to describe our methods precisely and in detail; we ask the Reviewer to approve that this sentence be retained in the manuscript.

  1.  

L123-124 Why a pair of "/" in  "/Eurofins Genomics/"

Answer:

We thank the Reviewer for this observation; we corrected the slashes to parentheses.

  1.  

L133 Actually some loci with NA's higher that 0.2 were, in fact, included in further analysis, as indicated at lines 179-180. Please, change this to something like "loci showing significantly different NA proportions were excluded from analysis". Also, indicate here the chosen significance threshold and by which program this was calculated.

Answer:

The sentence was rephrased in the revised manuscript. As we mentioned in the results the problematic loci (i.e. loci with high rate of null alleles) were observed only in few populations not all populations. Hence, we didn't remove these loci from further analysis. Finally, as we mentioned in the previous line, we estimated the frequency of null alleles using FreeNA. 

Revised sentence: “The microsatellite loci showing significantly high rates of NA (r > 0.2) were excluded from further analysis”.

  1.  

L149 The link (http://bioinformatics.org/tryphon/populations) is down. Please provide a valid one.

Answer:

The correct link was added.

  1.  

L159 A table detailing to which geographical location group belongs each strain would be a nice addition, perhaps adding a column to table 1.

Answer:

We thank the Reviewer for the suggestion, which indeed increases the lucidity of our manuscript. We inserted a new column entitled „Sample collection according to hatcheries” into Table 1, and drew up a map thereof (Figure 1).

  1.  

L163 Actually, when running structure, I always recommend testing more K values than a priori populations to check the possibility of hidden genetic sub-populations within the studied metapopulation.

Answer:

Thanks for your helpful suggestion. As you mentioned, hidden genetic sub-populations within the studied meta-population may be detectable in higher values of K. Hence, before selecting the final model parameters to run STRUCTURE, we ran STRUCTURE with difference ranges of K (k=1-20), but we didn’t find higher probabilities of Ln P (X/K) for K> 14. In all runs, the best k was obtained for K=14 or K=2. Hence, we ran the final model with k=1=14.

  1.  

L184 Please, when talking about mean alleles, do so in whole numbers; There is no way that any animal has a tenth of an allele.

Answer:

We thank the Reviewer for his/her observation; correction was made.

  1.  

L230 Actually, showing/discussing the second-best K (K=2), even in one liner, could be also illustrative. Please, consider this possibility.

Answer:

We thank the Reviewer for his/her observation; correction was made.
